# Resveratrol and Its Human Metabolites—Effects on Metabolic Health and Obesity

**DOI:** 10.3390/nu11010143

**Published:** 2019-01-11

**Authors:** Margherita Springer, Sofia Moco

**Affiliations:** 1Nestle Institute of Health Sciences, Nestle Research, EPFL Innovation Park, Building H, 1015 Lausanne, Switzerland; margherita.springer@rd.nestle.com; 2TUM Graduate School, Technical University of Munich, 85748 Munich, Germany

**Keywords:** resveratrol, polyphenols, metabolism, obesity, diabetes, metabolic pathways

## Abstract

Resveratrol is one of the most widely studied polyphenols and it has been assigned a plethora of metabolic effects with potential health benefits. Given its low bioavailability and extensive metabolism, clinical studies using resveratrol have not always replicated in vitro observations. In this review, we discuss human metabolism and biotransformation of resveratrol, and reported molecular mechanisms of action, within the context of metabolic health and obesity. Resveratrol has been described as mimicking caloric restriction, leading to improved exercise performance and insulin sensitivity (increasing energy expenditure), as well as having a body fat-lowering effect by inhibiting adipogenesis, and increasing lipid mobilization in adipose tissue. These multi-organ effects place resveratrol as an anti-obesity bioactive of potential therapeutic use.

## 1. Introduction

Resveratrol (3,5,4′-trihydroxy-*trans*-stilbene, RSV, Figure 1.1) is one of the most widely studied polyphenols with over ten thousand reports in the literature. This stilbene has attracted interest in popular culture over the years for its potential, yet often controversial, health benefits. RSV was first discovered in the roots of the white hellebore (*Veratrum grandiflorum* Loes. fil.) in 1939 [1], even though it is mostly recognized as the phytoalexin present in red wine [2]. When epidemiological studies showed the cardioprotective benefits of wine [3,4], the association with RSV followed [5], opening the field to a wealth of scientific research. RSV has since been identified as: being cancer chemoprotective [6], being anti-inflammatory [7], improving vascular function [8], extending the lifespan and ameliorating aging-related phenotypes [9,10], opposing the effects of a high calorie diet [11], mimicking the effects of calorie restriction [12], and improving cellular function and metabolic health in general [13].

Even though RSV has been widely studied both in vitro and in vivo, its mechanism of action across conditions and doses remains elusive. From the many effects elucidated in in vitro studies, most have failed to reproduce in vivo [14,15]. Reasons for such non-reproducibility among studies are diverse. One reason is its pharmacokinetics, as in humans RSV is highly absorbed orally (~70%), yet has poor systemic bioavailability (~0.5%) [16]. Rapid metabolism into RSV sulfate and glucuronide conjugates occurs, in addition to accumulation in tissues, as detected in radioactive trials and other studies [16,17,18]. Furthermore, a wide range of inter-individual responses upon oral ingestion of RSV is known in humans and is a common feature of many food bioactives [19,20]. Gut microbiota [20,21] and genetic background [22,23], including enzyme regioselectivity [24], are some of the possible known sources of the variation in responses. In contrast, in vitro studies have described an array of mechanistic effects that generate controversy given the likely non-physiological concentrations used, as well as the omission of the contribution of RSV metabolites [25].

Obesity is rising worldwide and is mainly attributed to changes in lifestyle, including overconsumption of food and decreased physical activity [26]. When energy intake exceeds energy expenditure over prolonged periods, then an obesity phenotype can develop [27]. Obesity has major health effects, and increased BMI (body mass index) is a risk factor in the development of type 2 diabetes, cardiovascular disease, dyslipidemias, non-alcoholic fatty liver disease, gallstones, Alzheimer’s disease and even certain cancers [26,27]. To reduce obesity, lifestyle changes and therapies are aimed at reducing energy consumption or increasing energy expenditure, or both, and/or by managing its side effects. This can be achieved by altering diet and increasing exercise. However not all individuals respond to these lifestyle changes, leading to surgery or drug therapies. For instance, RSV was shown to improve exercise endurance by significantly increasing aerobic capacity and consumption of oxygen in the gastrocnemius muscle in mice. RSV treatment induced oxidative phosphorylation and mitochondrial biogenesis by activating the peroxisome proliferator-activated receptor γ coactivator (PGC1α) through nicotinamide dinucleotide (NAD)-dependent deacetylase sirtuin-1 (SIRT1) leading to improved insulin sensitivity [13]. In addition, RSV has body fat-lowering effects as demonstrated by its anti-lipolytic affect in human adipocytes [28], as well as by decreasing adipocyte size, increasing SIRT1 expression, lowering nuclear factor kappa-light-chain-enhancer of activated B cells (NF-kB) activation and improving insulin sensitivity in visceral white adipose tissue in mice [29].

Even though RSV has been widely studied and associated with many benefits, many open questions remain, such as: (i) the activity of RSV at the nanomolar range, or of its human metabolites at the low micromolar range; (ii) levels of accumulation of these in target tissues able to elicit a biological effect; (iii) given an oral dose which preferred organs will be target sites of activity in which conditions or diseases; (iv) in physiological conditions which and how many protein targets are modulated; (v) how do these effects reproduce among individuals and populations; and (vi), how can one modulate RSV therapeutically. In this review we specifically discuss the role of RSV metabolism to better understand the mechanism of action, with particular emphasis on its potential effects in managing metabolic health and obesity.

## 2. Human Metabolism of Resveratrol

Being a phytoalexin, RSV levels vary greatly among food sources, seasons and batches. Certain foods are naturally rich in RSV, such as wine, peanuts and selected teas; however, RSV content in dietary sources remains at the lower milligram range [30]. To dose higher levels of RSV, dietary supplements are available in the open market at recommended daily doses between high milligram and gram levels [31]. Once RSV enters the gastro-intestinal tract, it suffers rapid and extensive biotransformation, with distribution into various organs (Figure 2), leading to consequences for its bioavailability and activity.

### 2.1. RSV is Absorbed and Metabolized in Target Tissues

The major function of the intestine is to digest food, making nutrients available for energy, while preventing the uptake of potentially harmful compounds. Bioactive compounds such as RSV can be perceived by the intestine as xenobiotics and therefore cross the intestinal epithelium to the blood via a transcellular pathway [32]. This route takes place through the enterocytes in the small intestine. Enterocytes, also known as absorptive epithelial cells, are the first site of reported RSV metabolism after being internalized by either passive diffusion [33] or carrier-mediated transport [32]. Once RSV is absorbed into the enterocyte, like other xenobiotics, it undergoes phase II of drug metabolism, producing polar metabolites, with easier excretion in the body. Specifically, RSV undergoes conjugation with sulfate (mediated by sulfotransferases, SULTs) and with glucuronate (mediated by uridine 5′-diphospho-glucuronosyltransferases, UGTs).

Drug metabolism takes place in multiple organs and cell types, and the observed biotransformation differs in metabolite levels [18], enzyme expression [34] and selectivity [24]. The superfamily of SULTs sulfates a broad spectrum of diverse endogenous and exogenous substrates. SULT1A1 is the main enzyme responsible for the transfer of a sulfate group to a hydroxyl group in phenolic compounds [35]. Biochemical studies have shown that, SULT1A1 is the main SULT responsible for the sulfation of RSV into RSV-3-*O*-sulfate (Figure 1.2), and to a minor extent SULT1A2, SULT1A3 and SULT1E1, whilst RSV-4’*O*-sulfate (Figure 1.3) is mainly produced by SULT1A2, and RSV-3,4’-*O*-disulfate (Figure 1.4) is mainly catalyzed by SULT1A2 and SULT1A3 [18,36], Table 1. Similar to the SULT family, UGT is also a large family of related enzymes involved in detoxification, which glucuronidate various substrates [37]. The glucuronidation of RSV is mainly catalyzed by UGT1A1 and UGT1A9, and to a minor extent by UGT1A6, UGT1A7, and UGT1A10 leading to RSV-3-*O*-glucuronide (Figure 1.5) and/or RSV-4’-*O*-glucuronide (Figure 1.6) [18], Table 1. In human tissues, the small intestine contains the highest amount of SULT proteins of any tissue, yet preferably expresses SULT1B1, followed by SULT1A3 and then SULT1A1 [34]. In the liver and kidney, however, SULT1A1 is the main SULT protein isoform expressed [34]. SULT1A1 was also recently found in adipocytes, and modulates RSV sulfation in the SGBS (Simpson-Golabi-Behmel syndrome) human adipocyte [28]. Again, the intestine is the human tissue with higher UGT1A content, largely expressing UGT1A10 and 1A1, while the kidney preferably expresses UGT1A9, and the liver expresses UGT1A4, 1A1, 1A6 and 1A9 at the protein level [38]. Of note is the inter-species variation of phase II metabolism in which RSV sulfates are the main conjugates in humans, while glucuronides are the preferred conjugates in pigs and rats [32]. More importantly, drug metabolism is a known cause of inter-individual variability, as both SULTs and UGTs have genetic polymorphisms [22,35]. From animal and human studies, RSV and metabolites thereof have been reported to reach many tissues and biofluids, Table 1.

After absorption and conjugation, RSV sulfates and glucuronides have two fates: they can either be transported through the apical membrane and reach the intestinal lumen or they can pass through the basolateral membrane and enter the bloodstream (Figure 2A). On both membranes, the enterocyte contains ABC (ATP-binding cassette) transporters, which are part of a large family of transport proteins and are considered to be instrumental in drug absorption and response [39]. On the apical side, breast cancer resistance protein (BCRP/ABCG2), multidrug resistance-associated protein 2 (MRP2/ABCC2), and P-glycoprotein (P-gp/MDR1/ABCB1) transporters are expressed, while on the basolateral side, MRP3 (ABCC3) is expressed instead. BCRP and MRP2 play a major role in the efflux of conjugated RSV, while P-gp plays a minor role. On the basolateral membrane, conjugated RSV is transported into blood capillaries by the ABC transporter MRP3 [32]. Transporters are not limited to playing a role in the absorption and distribution of RSV and metabolites in the small intestine, as they are also expressed in other tissues, such as the liver and kidneys [40]. When RSV and metabolites reach the bloodstream, they can be transported by binding to blood proteins such as lipoproteins [5], hemoglobin, and albumin, before reaching other tissues, such as liver, kidney and other peripheral tissues.

### 2.2. The Gut Microbiome Metabolizes RSV and RSV Influences Gut Microbial Composition

RSV and the metabolites thereof can be further metabolized in the colon by the gut microbiota (Figure 2B). Here, RSV metabolites may be hydrolyzed, regenerating RSV, and additional reduction reactions may take place. The most described microbial metabolite of RSV is dihydroresveratrol (DHR, Figure 1.7, Table 1). Intestinal bacteria are able to metabolize RSV into DHR by reduction of the double bond between the two phenol rings. DHR produced by the intestinal bacteria can then be absorbed, conjugated and excreted in the urine. In addition to DHR, 3,4’-dihydroxy-*trans*-stilbene (Figure 1.8, Table 1) and lunularin (Figure 1.9, Table 1) have also been identified as gut metabolites of RSV in human urine. A large inter-individual variation between subjects was observed, in which some proved to be lunularin producers, DHR producers or mixed producers, according to levels of these metabolites [20]. Using 16s rRNA sequencing of fecal samples, lunularin producers were associated with a higher abundance of *Bacteroidetes, Actinobacteria, Verrucomicrobia*, and *Cyanobacteria* and a lower abundance of *Firmicutes* than either the DHR or mixed producers. The bacterial strains *Slackia equolifaciens* and *Adlercreutzia equolifaciens*, species not previously known to metabolize RSV, were found to metabolize RSV to DHR [20].

From the gut, RSV microbial metabolites may be absorbed and reach the liver as well as other tissues for further metabolism or excretion. A common feature of certain xenobiotics, including RSV, is the enterohepatic circulation, in which RSV metabolites may go from the liver to the bile and re-enter the intestine. From the small intestine, RSV and metabolites may suffer hydrolysis before reaching the portal circulation and being re-transported into the liver. The extensive presence of RSV and metabolites in the bloodstream can be attributed to enterohepatic circulation [16], Figure 2B.

Beyond the metabolic capacity of the gut microbiota to convert polyphenols such as RSV into often smaller and simpler molecules, the gut microbiota has been associated with other functions. The influence of gut microbiota on the metabolism of polyphenols, and conversely the modulation of the gut microbial composition due to polyphenol intake, are significant topics for understanding the metabolism and activity of these bioactives in humans [19,50]. RSV supplementation is known to alter the microbiome in at least two ways, by acting as an antimicrobial agent and through modulating gut microbial composition. Dao et al. reported that RSV-treated mice lacked three gut bacteria compared to controls: *Parabacteroides jonsonii*, *Alistipes putredinis* and *Bacteroides vulgatus* [51]. RSV increased levels of *Bifidobacterium* and *Lactobacillus*, while decreasing levels of *Escherichia coli* and Enterobacteria in RSV-treated rats [52]. The antimicrobial activity of RSV against *E. coli* lies in the inhibition of bacterial cell growth by suppressing FtsZ (filamenting temperature-sensitive mutant Z) expression and Z-ring formation, essential for cell division [53]. RSV has a wide range of antimicrobial activity, as it seems to be effective against both gram-positive and gram-negative pathogenic bacteria [54].

RSV supplementation can alter the gut microbial composition and some suggest this may be an essential mechanism of action of RSV [55]. RSV leads to functional changes in the gut microbiome of obese mice, including: decreased relative abundance of *Turicibacteraceae*, *Moryella*, *Lachnospiraceae*, and *Akkermansia* and increased relative abundance of *Bacteroides* and *Parabacteroides*. Glucose homeostasis in obese mice was improved by faecal transplantation from healthy RSV-fed donor mice [55]. Obese gut microbiota has been associated with a reduced *Bacteroidetes/Firmicutes* ratio in mice and humans [56] and RSV was found to increase this ratio in rodent studies [55,57]. *Firmicutes*, more prevalent in the obese, produce greater amounts of energy from dietary fiber than other major gut bacterial phyla, such as *Bacteroidetes*, by increasing the production of short chain fatty acids (SCFAs) [58]. Furthermore, an increase in the *Bacteriodetes* population in the gut was also observed in overweight men, upon RSV and epigallocatechin-3-gallate supplementation [59]. *Bacteroidetes* is also associated with postprandial fat oxidation [58]. In addition to SCFAs production, the gut microbiota produces many other small molecules. Dietary choline, l-carnitine and lecithin can be converted into trimethylamine in the gut, which is converted to trimethylamine-*N*-oxide (TMAO) in the liver. TMAO production is associated with chronic diseases like cardiovascular disease, type II diabetes, and obesity [58,60]. RSV supplementation was shown to reduce TMAO production by increasing the *Bacteroidetes* population in the gut of mice [61].

### 2.3. Biotransformation of RSV Limits Plasma Bioavailability

The metabolic fate of RSV in the body is therefore widespread into different tissues, and its metabolism is rapid and extensive. A preclinical study in rats demonstrated that only a small fraction of RSV (1.5%) is able to escape conjugation and enter the bloodstream unmodified. About 75% enters the enterocyte while the remaining 25% is directly excreted. Once inside the cell, 60% is glucuronidated and 13.5% sulfated. These conjugates partially return to the intestine (42% glucuronides and 12% sulfates), leaving 17% glucuronides and 1.5% sulfates in the bloodstream [32,62]. Administering the metabolites RSV-3-*O*-sulfate (Figure 1.2) and RSV-4’-*O*-sulfate (Figure 1.3) to mice has shown that these metabolites are absorbed yet at low bioavailability (14% and 3%, respectively). Interestingly, regeneration of free RSV (2%) into the bloodstream was observed, indicating in vivo hydrolysis of sulfates, depending on membrane transporter activities [63].

Many studies, both preclinical and clinical, have detected RSV metabolites in plasma (Table 1). Plasma concentration is an indicator of RSV bioavailability and determines the amount of RSV and metabolites available to peripheral target tissues. Regarding human studies, plasma concentration of RSV after single (Table 2) and repeated dosing (Table 3) was measured for studies from 2010 to 2018. Earlier studies have been reviewed already by Cottart et al. [64]. Administered in either single or repeated dosing, the peak levels of RSV in plasma were very low, given its poor bioavailability.

## 3. Molecular Action of Resveratrol and Metabolites

### 3.1. RSV Modulates a Panoply of Protein Targets

Many and diverse effects have been described for RSV, indicating an array of possible protein targets that can be (in)directly modulated by this compound [75,76,77,78,79,80]. Recently, a computational approach was used to map all publically available polyphenol-protein interactions [81]. Among all polyphenols and human metabolites, RSV was found to be one of the polyphenols with the most known interactions with proteins (738 RSV-protein interactions). Only five protein interactions were reported with RSV metabolites, specifically interacting with DHR, highlighting the lack of studies on RSV metabolites. Taken together, the protein interactome of RSV and DHR led to 743 interactions (Figure 3). The interacting proteins can be classified in terms of diseases, using DAVID’s [82] genetic association database (GAD) [83] disease classification system. RSV showed low coverage for most diseases (<50%), yet a widespread representation, highlighting its potential pleotropic effect. RSV modulates genes within pathways of cancer, metabolic and cardiovascular diseases, and to lesser extent other disease classes (Figure 3A). Taking the same list of RSV-interacting proteins, these could be classified in terms of protein super-families, using the InterPro [84] protein classification in DAVID. This analysis highlighted an enrichment in many protein super-families modulated by RSV (Figure 3B), of relevance in metabolic diseases and obesity, such as nuclear hormone receptor-type (e.g., PPARγ), insulin related, NF-κB, enolases, sirtuins, and nitric oxide related proteins. Using STITCH [85], a protein-metabolite database, proteins of experimental evidence were further selected to interact with RSV (Figure 3C). This obtained network highlighted, with substantial overlap, the enrichment of RSV-interacting proteins previously identified by InterPro classification (Figure 3B).

While RSV may establish a large number of possible interactions, some of these proteins are already established direct targets of RSV, at least in vitro [77]. The structure of some of these protein-RSV complexes can be found in the Protein Databank (PDB) [86], accounting for 24 RSV-protein complexes, and even a handful of RSV metabolite protein complexes.

### 3.2. RSV Increases Energy Expenditure and Vascular Function

One of the most studied mechanisms of RSV is its capacity to increase energy expenditure by modulating protein targets within central energy pathways and signaling, specifically by inducing mitochondrial biogenesis. RSV can directly activate SIRT1 and SIRT5. Because sirtuins are NAD-dependent deacetylases, they directly depend on NAD^+^ and therefore are quite sensitive to cellular energy via imbalances of the redox pair NAD^+^/NADH. Sirtuins act as caloric restriction mimetics, with potential benefits in longevity and preventing age-related complications, as well as type II diabetes and obesity. By activating SIRT1, RSV elicits deacetylation of PGC1α, a key regulator of energy metabolism, leading to decreased glycolysis in muscle and the liver, and increased lipid use [13,87]. In addition, RSV inhibits ATP production by interfering with mitochondrial function, leading to an increase of AMP/ATP ratio, which activates AMP-activated protein kinase (AMPK) [75]. AMPK is a pivotal protein in governing energy homeostasis and its activation takes place in cases of nutrient starvation or in the presence of agonists, such as certain drugs (e.g., metformin) or natural compounds such as RSV. Furthermore, AMPK may inhibit mTOR signaling, that in certain species has been associated with an extended lifespan, given its anti-ageing effects. The signaling pathway of AMPK crosstalks with Akt (protein kinase B). Akt are kinases involved in metabolism and cell proliferation and are part of the PI3K/AKT/mTOR pathway that governs the cell cycle. Activation of Akt reduces the activity of AMPK through direct phosphorylation [79].

Interestingly, AMPK and Akt have been shown to directly phosphorylate the endothelial nitric oxide synthase (eNOS). eNOS, responsible for the production of nitric oxide (NO), is activated by shear stress and agonists, and has a protective function in the cardiovascular system. RSV has been shown to be such an agonist, by improving vascular function and vasoprotective effects, including vascular NO production and bioavailability, and perivascular adipose tissue function [88]. Akt and AMPK may contribute to the stimulation of NO production by eNOS in response to RSV treatment [79].

Beyond central metabolism and bioenergetics, RSV also impacts lipid metabolism. RSV can directly inhibit PPARγ, a nuclear receptor expressed in adipose tissue, or do so indirectly via SIRT1, leading to decreased adipogenesis and increased lipolysis [87]. SIRT1 is also known to repress NF-κB activity, and thereby reduce inflammation. RSV modulates inflammation by directly interacting with cyclooxygenases (COX), which catalyze the formation of prostaglandins, bioactive lipids with hormone-like effects [77].

### 3.3. Effect of RSV on Epigenietics

While RSV has an impact on metabolism, some of its mechanisms are a consequence of epigenetic modifications. In fact, RSV can induce epigenetic modifications to the DNA sequence. Epigenetic modifications include DNA methylation, histone modifications and nucleosome positioning. These modifications can interact with each other to influence gene expression [90]. RSV supplementation can influence DNA methylation and histone modification, with the latter being most relevant in the context of obesity and energy balance. Histones are subject to post-transcriptional modification like methylation, phosphorylation, ubiquitination, SUMOylation and ADP-ribosylation. These modifications can be reversed by methyltransferases, histone demethylases, kinases, histone acetyltransferases and histone deacetylases. For instance, RSV can influence histone deacetylation via sirtuins, and sirtuins activated by RSV can deacetylate sites on PGC1α [13]. The nuclear bile acid receptor farnesoid X receptor (FXR) is a target a SIRT1 that plays a critical role in the regulation of lipid and glucose metabolism. RSV treatment reduced acetylated FXR levels, with benefits for metabolic health [91].

### 3.4. RSV Influences Redox Metabolism

RSV has 16-times lower antioxidant capacity than alpha-tocopherol [92], nature’s ubiquitous antioxidant, and therefore will be inefficient per se for radical scavenging in physiological conditions. However RSV, in addition to other polyphenols, has been described as undergoing redox cycling, being able to adopt a quinone-like structure and generate reactive oxygen species (ROS). ROS production leads to the activation of the nuclear factor-erythroid 2-related factor-2 (Nrf2), a transcription factor that regulates redox status, and reacts against stresses. In addition, Nrf2 improves cellular recycling and cross-talks with central and lipid metabolism, as well as modulates phase I and II metabolism enzymes and transporters [93,94]. One of these phase II detoxifying enzymes, quinone reductase 2 (QR2) has been shown to interact directly with RSV. The inhibition of QR2 by RSV may induce other cellular antioxidant enzymes and increase cellular resistance to oxidative stress [77]. Oxidative stress contributes to type II diabetes, and RSV showed antioxidant effects after eight weeks of supplementation with 800 mg/day in the blood and PBMCs (peripheral blood mononuclear cells) of diabetic patients. After RSV consumption, the expression of Nrf2 and superoxide dismutase (SOD) was significantly increased, along with reductions in body weight, BMI and blood pressure [95].

### 3.5. RSV Metabolites Exhibit Activity

RSV metabolites have been examined for their potential activity only recently. A biochemical study compared the action of RSV, and RSV-3- and 4’-*O*-sulfates (Figure 1.2,3) on three direct targets: COX, SIRT1 and QR2. RSV and metabolites can inhibit both COX and QR2 enzymes. SIRT1 is activated by RSV and metabolites, but the activation seems to be a substrate-dependent phenomenon questioning in vivo relevance [96]. Comparable in vitro activities were also found for RSV glucuronides [97]. The ability to bind to human serum albumin was found to be comparable between RSV, RSV-4’-glucuronide (Figure 1.6) and DHR (Figure 1.7), while RSV-3-*O*-glucuronide (Figure 1.5) showed a slightly lower affinity. RSV-4’-glucuronide was able to inhibit COX-2 and DHR presented comparable activity in inhibiting NO production [97]. The metabolites RSV-3- and 4’-*O*-sulfates were also studied in vivo, by dosing these conjugates directly to mice. Both compounds led to low bioavailability, but hydrolysis of sulfate moieties was identified, contributing to the recirculation of free RSV. In human cancer cells, RSV metabolites partially regenerated free RSV and prompted autophagy and senescence [63]. Thus, these studies lead to the suggestion of a metabolic interplay between RSV and phase II metabolites in the body. According to the cell’s specific conditions, RSV metabolites function as a pool of RSV, actively contributing to a wide variety of actions, previously solely attributed to RSV.

## 4. Resveratrol, Metabolic Health and Obesity

Obesity is characterized by an excess accumulation of adipose tissue which is a risk factor for the development of chronic diseases [26,98]. Adipose tissue is composed of specialized cells, adipocytes, which store and release energy. To store energy, adipocytes convert free fatty acids to triglycerides through lipogenesis, and to release energy triglycerides are metabolized through a process called lipolysis. Adipocytes also produce hormones, adipokines, which relay information from the adipose tissue to the central nervous system [99]. Reduction of adipose tissue through increased physical activity and decreased energy intake can reduce the risk of adverse health outcomes. Though increased exercise and reduced calorie intake are effective methods to reduce adiposity, compliance is low and genetic factors may be unfavorable; therefore, alternative treatments are needed. RSV has been shown to influence adipose tissue function [100].

RSV was found to have an anti-lipolytic effect at low physiological concentration in a human adipocyte model [28]. A decreased sulfation of RSV, by knock-down of SULT1A1, resulted in an increased anti-lipolytic effect, as demonstrated by lower glycerol accumulation, probably attributed to lower activity of the lipolytic protein perilipin, suggesting the role of phase II enzymes in RSV bioavailability in adipose tissue. A comprehensive review on effects of RSV on adipose tissue [101] detailed effects on how RSV can modulate adipogenesis, apoptosis, de novo lipogenesis and lipoprotein lipase functions, lipolysis, thermogenesis, and fatty acid oxidation, in vitro and in rodent models.

Although adipose tissue is an attractive target site of RSV, only a few human studies have so far been conducted to consider RSV-treatment in this tissue. A recent human clinical trial investigated the effect of RSV supplementation on the adipose tissue metabolome [102]. In this study, male subjects with metabolic syndrome (characterized by elevations in at least three of the following: abdominal obesity, blood lipids, blood pressure and fasting blood glucose) were treated for four months with 1 g of resveratrol. The metabolome of these biological samples were characterized using untargeted metabolomics. This approach identified 282 metabolites in adipose tissue, of which 45 changed significantly in response to RSV treatment. RSV supplementation was associated with increased long chain-fatty acids, increased polyunsaturated fatty acids and decreased steroids [102]. Another study in obese men, supplemented for 30 days with 150 mg of RSV, observed changes in adipose tissue morphology. RSV decreased abdominal subcutaneous adipocyte size. Transcriptome profiling on the adipose tissue samples and subsequent pathway analysis identified an enrichment of genes involved in cell cycle regulation pathways, suggesting enhanced adipogenesis [103].

In terms of obesity and the sphere of weight management, RSV has demonstrated significant improvement of glucose control and insulin sensitivity in diabetics [104]. A few studies indicate potential in enhancing adipogenesis as well as lipid markers in adipose tissue [102,103], therefore this is an application area worth further exploring. Even though clinical interventions can be particularly challenging, perhaps it is in the preventive health space that RSV can offer its full potential [31,78].

## 5. Outlook

With over 140 human clinical trials using RSV (clinicaltrials.gov), and more than 10,000 scientific publications describing the uses and effects of RSV, much research has been conducted on this small molecule. Due to varying doses, disparate experimental setups, low statistical power, and a myriad of biological or other types of confounders, the ultimate fate and effect of RSV in humans remains elusive. Mechanisms of action are varied and of potential benefit for cardiovascular health, obesity, metabolic health, inflammation, and cancer management, therefore, RSV is of wide pleiotropy. Systems-driven approaches [81] aid in mapping effects of multi-targeted compounds such as RSV and highlight links between bioenergetics [27], phase II metabolism [38] and redox pathways [93], linked by protein targets modulated by RSV.

The bioavailability of RSV is often mentioned as a limitation and is subject to controversy. Nevertheless, studies on RSV metabolites seem to be encouraging, as these either have similar effects or can act as a RSV pool in the body, fostering the metabolic effects previously solely attributed to free RSV [63]. On the note of enhancing knowledge on RSV metabolites and its potential actions, the use of untargeted metabolomics analysis [105] can widen the spectrum of RSV metabolites known so far, and also map metabolic sub-network effects induced by RSV [102]. Stable isotopes are an elegant tool to unravel metabolic fate of specific compounds [106] and offer advantages compared to radioactive labeling strategies.

Inter-individual variability [19] upon RSV intake can be attributed to various factors such as: (i) gut microbiota composition; (ii) genetic polymorphisms in phase II metabolism enzymes (e.g., UGTs, SULTs) and transporters, including tissue specificity and/or enzymatic regioselectivity; (iii) variability inherent in specific ethnicities or geographic subpopulations; (iv) specific lifestyles and diets; or (v) simply part of natural human variation. As bioactive-intervention studies often rely on small human studies, the variability can be overpowering. These factors need to be taken into account in future studies. Efforts to conduct larger and more deeply characterized studies could be an aim of the research community in order to bridge the current gaps in knowledge on RSV metabolism and beneficial effects.

## Figures and Tables

**Figure 1 nutrients-11-00143-f001:**
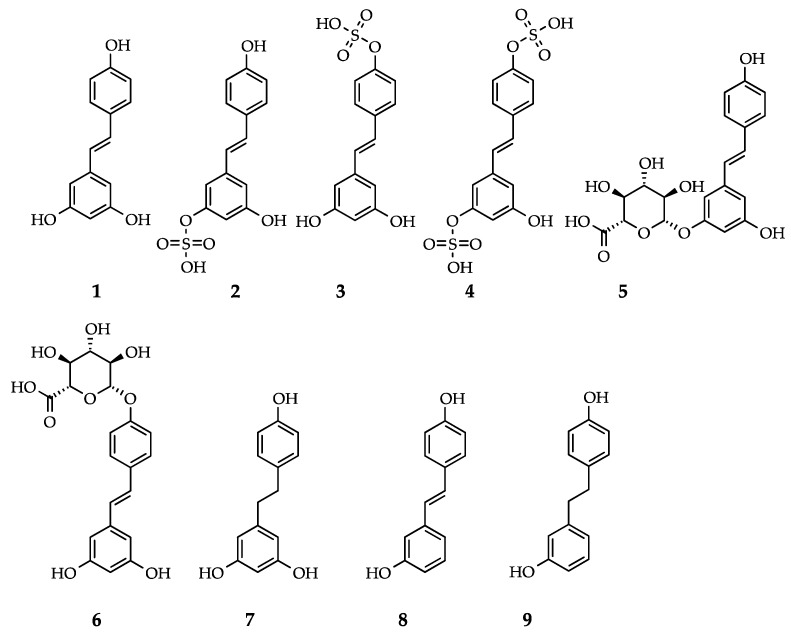
Resveratrol and reported human metabolites: (**1**) *trans*-resveratrol (RSV); (**2**) *trans*-resveratrol-3-*O*-sulfate; (**3**) *trans*-resveratrol-4’-*O*-sulfate; (**4**) *trans*-resveratrol-3,4’-*O*-disulfate; (**5**) *trans*-resveratrol-3-*O*-glucuronide; (**6**) *trans*-resveratrol-4’-*O*-glucuronide; (**7**) dihydroresveratrol (DHR); (**8**) 3,4’-*O*-dihydroxy-*trans*-stilbene; and (**9**) lunularin (also see Table 1).

**Figure 2 nutrients-11-00143-f002:**
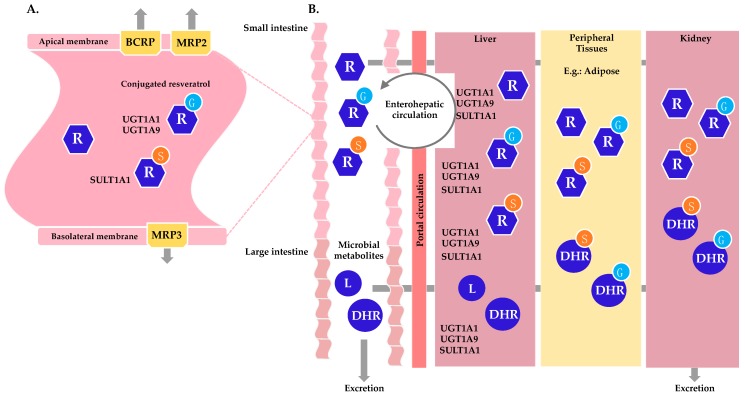
Metabolic fate and biotransformation of resveratrol in the human gastro-intestinal tract, and metabolism in different organs. (**A**) Metabolism of resveratrol (R) in the small intestine’s enterocyte. Resveratrol is absorbed into the enterocyte and undergoes sulfation (S) by SULT1A1 and glucuronidation (G) by UGT1A1 and UGTA9. Conjugated resveratrol exits the cell via BRCP and MRP2 transporters on the apical membrane and MRP3 on the basolateral membrane. A small faction of resveratrol escapes conjugation and exits the enterocyte via the basolateral membrane. (**B**) Integrated human metabolism of resveratrol. Resveratrol and conjugated metabolites exit the apical membrane of the small intestine and move towards the large intestine where they can be metabolized by the gut microbiota to generate dihydroresveratrol (DHR), lunularin (L) and 3,4’-dihydroxy-*trans*-stilbene (not shown). Resveratrol and metabolites that exit the enterocyte enter portal circulation. The liver expresses SULT1A1, UGT1A1 and UGTA9, which can further conjugate resveratrol. In addition, conjugated resveratrol and metabolites undergo enterohepatic circulation, leaving the liver to be reabsorbed in the intestine after hydrolysis, and entering portal circulation to reach the liver again for further metabolism. From the liver, resveratrol and metabolites enter systemic circulation and are absorbed by peripheral tissues, such as adipose tissue. The kidneys also participate in the metabolism of resveratrol, leading to excretion of polar resveratrol metabolites.

**Figure 3 nutrients-11-00143-f003:**
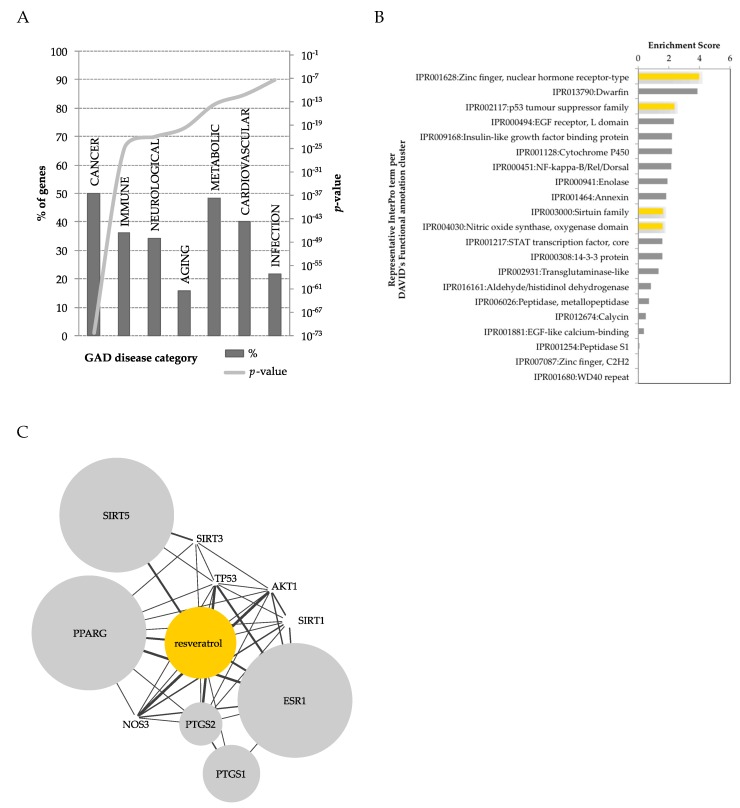
Using Lacroix et al.’s [81] human polyphenol-protein interactome, proteins (743) interacting with resveratrol and dihydroresveratrol were functionally analyzed in DAVID [82]. (**A**) Resveratrol interacting genes annotated according to the genetic association database (GAD) [83] disease categories, according to percentage of genes and *p*-value. (**B**) Clustering of protein super-families according to the InterPro [84] protein classification, at the highest stringency (in yellow, protein classes represented in (**C**). (**C**), Human resveratrol-protein interactome using experimental evidence obtained from STITCH [89]. The size of the node reflects the strength of the resveratrol-protein interaction. Depicted proteins: peroxisome proliferator-activated receptor gamma (PPARG), NAD-dependent deacetylase sirtuin-1 (SIRT1), 3 (SIRT3) and 5 (SIRT5), nitric oxide synthase 3 (NOS3), prostaglandin-endoperoxide synthase 1 (PTGS1) and 2 (PTGS2), estrogen receptor 1 (ESR1), tumor protein p53 (TP53), and AKT serine/threonine kinase 1 (AKT1).

**Table 1 nutrients-11-00143-t001:** Human, rat, and mouse resveratrol metabolites after oral administration in different biofluids and tissues (see structures in Figure 1).

Metabolite	Species and Tissue or Biofluid [Reference]
*trans*-resveratrol	Human: serum [41], plasma [15,16,42], urine [16,20] Rat: plasma [43,44,45], liver [44], lung [44], brain [44], kidney [44] Mouse: plasma [18,44,46], liver [18,44,46], lung [18,44], brain [44,46], kidney [18,44], heart [18,46], stomach [18], duodenum [18], intestine [18], muscle [18], spleen [18], thymus [18], urine [18], feces [18]
*trans*-resveratrol-4’-*O*-glucuronide	Human: serum [41], plasma [42], urine [42] Mouse: plasma [46]
*trans*-resveratrol-3-*O*-glucuronide	Human: serum [41], plasma [42], *urine* [16,42] Rat: plasma [43], liver [47], adipose tissue [47,48], skeletal muscle [47] Mouse: plasma [18,46], liver [18,46], lung [18], brain [46], kidney [18], heart [18,46], stomach [18], duodenum [18], intestine [18], muscle [18], spleen [18], thymus [18], urine [18], feces [18]
*trans*-resveratrol-diglucuronide	Human: plasma [49], urine [49] Mouse: plasma [46], liver [46]
*trans*-resveratrol-3-*O*-sulfate	Human: plasma [42], *plasma* [16], *urine* [16] Rat: adipose tissue [47,48] Mouse: plasma [18,46], liver [18,46], lung [18], brain [46], kidney [18], heart [18,46], stomach [18], duodenum [18], intestine [18], muscle [18], spleen [18], thymus [18], urine [18], feces [18]
*trans*-resveratrol-4’-*O*-sulfate	Human: plasma [42], *plasma* [16], *urine* [16,42] Rat: liver [47], adipose tissue [47,48]
*cis*-resveratrol-3-*O*-sulfate	Rat: adipose tissue [47,48]
*tran**s*-resveratrol-3,4’-disulfate	Human: *plasma* [42] Rat: adipose tissue [48] Mouse: plasma [18], liver [18], lung [18], kidney [18], heart [18], stomach [18], duodenum [18], intestine [18], muscle [18], urine [18], feces [18]
*trans*-resveratrol-glucuronide-sulfate	Mouse: plasma [46], liver [46]
dihydroresveratrol	Human: urine [20], plasma [15] Rat: liver [47], skeletal muscle [47]
dihydroresveratrol-glucuronide	Human: urine [16] Rat: liver [47] Mouse: plasma [46], liver [46]
dihydroresveratrol-sulfate	Human: urine [16] Rat: liver [47], adipose tissue [47] Mouse: plasma [46], liver [46]
dihydroresveratrol-glucuronide-sulfate	Mouse: plasma [46]
3,4’-dihydroxy-*trans*-stilbene	Human: urine [20]
lunularin	Human: urine [20]

*italic:* likely identification.

**Table 2 nutrients-11-00143-t002:** Reported resveratrol plasma concentration in humans after a single dose of resveratrol (studies after 2010).

Number of Participants, Characteristics	Dose (mg)	Administration	Peak Plasma Concentration (ng/mL)	Reference
15, healthy	500	Tablet	71.18	[65]
6, low BMI 6, high BMI	2125	Tablet and drink	634.32 498.56	[66]
7, healthy	500	Capsule ^1^	1598	[67]
8-9/dose, healthy	250 500	Capsule	5.65 14.4	[68]
2, healthy	146	Lozenge	328.5	[69]

body mass index (BMI); ^1^ Capsule also contained 10 mg of piperine.

**Table 3 nutrients-11-00143-t003:** Reported resveratrol plasma concentration in humans after repeated doses of resveratrol (studies after 2010).

Number of Participants, Characteristics, Study Type	Dose (mg/day)	Days	Administration	Peak Plasma Concentration (ng/mL)	Reference
6, low BMI 6, high BMI	2125	11	Tablet and drink	903.0 245.0	[66]
35, healthy males, cross-over study	800	5	Capsule Dairy drink Soy drink Protein-free drink	Capsule: 0.56 Dairy drink: 0.61 Soy drink: 0.58 Protein free drink: 0.70	[70]
7, healthy	500	28	Capsule ^1^	2967.25	[67]
40, healthy, repeated sequential dosing	500 1000 2500 5000	29	Caplet	43.8 141 331 967	[71]
6, patients with hepatic metastases, randomized double-blind clinical trial	5000	14	Micronized resveratrol mixed in liquid	1942	[72]
8, healthy subjects	2000	7	Capsule	1274	[73]
19, overweight or obese, randomized, double-blind, placebo-controlled, crossover intervention	30 90 270	6	Capsule	181.31 532.00 1232.16	[74]

body mass index (BMI); ^1^ Capsule also contained 10 mg of piperine.

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
