# Peer review of "Resveratrol and Its Human Metabolites—Effects on Metabolic Health and Obesity"

_nutrients, 2019, doi:10.3390/nu11010143_

Round 1

Reviewer 1 Report

The review “Resveratrol and its human metabolites – effects on metabolic health and obesity” by M. Springer and S. Moco reports recent findings about resveratrol and its metabolites in both in vitro and in vivo studies. The manuscript is well written and organized. The literature is also very comprehensive about the treated topic. Only very few minor English and editing errors are present. Tables and figures are well presented and summarize well major findings which are then analyzed in the main text.

Author Response

We thank the reviewer for her/his time to read and comment our manuscript. We believe we have improved the English of the whole manuscript throughout accordingly.

Reviewer 2 Report

Overall impression: A very nice manuscript that is generally well written (with a few exceptions noted below). It covers several topics that are highly relevant to the potential clinical use of RSV. It is readable and informative. 

Specific Comments:

Line 57: strange use of the word “recurring.” Better to say “leading to” 

Line 59: change “mice muscle” to specify which muscle was measured.

Lines 232-46:  The central theme of this paragraph is how RSV modulates energy expenditure. However, in lines 240-46 the overall intent of this section is not exactly clear. Its just a list of various things that stem from AMPK, then gets into AMPK and Akt interactions. Please describe a context for these changes that can relate back to central theme or the paragraph.

Liners 250-51: I suggest writing in this sentence that phosphorylation of eNOS by AMPK and Akt results in greater NO production. This will make the intent of paragraph more clear to the readers because phosphorylation of eNOS by kinases does not always lead to activation. For example eNOS phosphorylation by ERK actually reduces NO production.

Line 275-276: please specify cell type used in the cited study.

Line 326: the phrase “Being varying doses” does not make sense to me. Are the authors trying to say “Because of varying doses…?”

Lines 347-50: this section seems to fit better at the end of “Section 4. Resveratrol, metabolic health, and obesity.”

I think a sentence or two after line 346 that offers “future directions” as the authors see it, would be a nice addition to conclude the paper. Something that is complimentary to the last sentence (line 350-51).

Author Response

We thank the reviewer for her/his time to read and comment our manuscript.

As requested, we improved the manuscript by incorporating the reviewer’s suggestions and we tried to address her/his concerns.

Line 57 – the word was changed to ‘leading to’.

Line 59 – the muscle refers to gastrocnemius muscle from mice (Lagouge et al., 2006) and this was added accordingly.

Lines 232-46: Section 3 of the manuscript was changed and organized, as we introduced subsections. This particular section was re-written.

Lines 250-51: This particular section was re-written.

Line 275276: This was a biochemical study and the wording was then changed in the text.

Line 326: The wording was changed to ‘Due to varying doses’.

Lines 34750: This section was then placed in Section 4 as suggested.

I think a sentence or two after line 346 that offers “future directions” as the authors see it, would be a nice addition to conclude the paper. Something

that is complimentary to the last sentence (line 35051): A sentence was then added.

Reviewer 3 Report

Springer and Moco present for publication on Nutrients a review on Resveratrol.

Resveratrol is a very famous polyphenol and this review is definitely not the first one to be published on this subject. This reviewer appreciates the effort the authors put in preparing the manuscript and in writing it in an original way.

Despite the redundancy of the subject, the authors wrote a very interesting review. The manuscript addresses some interesting and unusual aspects of Resveratrol, that, overall, render the manuscript rare compared to other reviews on the same subjects. I appreciated a lot the focus on Resveratrol metabolites, as well as on their destiny and intracellular function. This aspect was well reviewed and makes this manuscript novel and interesting. The style used to write is friendly. This reviewer thinks that the broad readership of Nutrients will find in this review fresh stimuli and new concepts, as well as milestones of the field easily explained by two experts.

Despite my positive comments, I think that the review must be improved and that it requires a revision. The review is brilliant till Section 2 . Starting from Section 3, the review starts to be simplistic and chaotic. I think that too many concepts were squeezed together and this results in a  second half of the manuscript that lost that didactic aspect that a review should always have.

I thus request some compulsory changes before the manuscript can be further considered for publication

In section 2 the authors are too simplistic in addressing the subjects “influence of gut microbiota on reseveratrol metabolites “ and “influence of resveratrol on gut microbiota”. The authors conclude at line 168 that further research on these aspects is expected to follow. This is unacceptable. This reviewer is sure that there are plenty of brilliant manuscripts that are approaching these aspects . These cannot be merely summarize with a sentence like “research is expected to follow”.

My first request is thus to introduce a sub-section where the authors describe the effect of resveratrol and its metabolites on gut microbial composition (keywords: resveratrol as antibiotic or prebiotic; selectivity of resveratrol for specific bacterial strains; modulation of TMA production; the effect on gut yeast species and effect on Butyrate producing bacteria…..)

Second request: this reviewer finds the subject epigenetic effect of resveratrol consumption not addressed at all. The authors should address this point in a subsection the revised version of the review

Third request: Section 3 is way too confusing. It is discouraging and suggests a promiscuous affinity of resveratrol for proteins. Would it be better to start discussing proteins that were co-crystallized with resveratrol or for which NMR data in complex with resveratrol are available (RCSB website)? Apart from these surely interacting proteins ( proved physical interaction between resveratrol and proteins) the rest can be just defined as proteins directly or indirectly “modulated” by resveratrol. This reviewer request to reorganize section 3 following the above mentioned indications.

At line 320 the authors cite the Wnt pathway. This pathway is coming out of the blue and makes the sentence confusing. I agree with the importance of referring to this pivotal intracellular pathway, notwithstanding, the author should either better explain it with few sentences or simply remove the full  sentence. 

Author Response

We thank the reviewer for her/his time to read and comment our manuscript.

As requested, we improved the manuscript by incorporating the reviewer’s suggestions and we tried to address her/his concerns.

We thank the reviewer for her/his time to read and comment our manuscript.

As requested, we improved the manuscript by incorporating the reviewer’s suggestions and we tried to address her/his concerns.

First request

A section was included on gut microbiota and resveratrol (2.2)

Second request

A section was included on epigenetics and resveratrol (3.3), even if epigenetic events were mentioned throughout section 3.

Third request:

Section 3 was completely re-organized and re-written. Subsections introduced, so we hope to readability was also improved.

At line 320 the authors cite the Wnt pathway:

This was removed, as suggested.

Round 2

Reviewer 3 Report

The revised version of the review can be accepted for publication